# Antimicrobial Resistance, Virulence Gene Profiling, and Spa Typing of *Staphylococcus aureus* Isolated from Retail Chicken Meat in Alabama, USA

**DOI:** 10.3390/pathogens14020107

**Published:** 2025-01-22

**Authors:** Rawah Faraj, Hazem Ramadan, Kingsley E. Bentum, Bilal Alkaraghulli, Yilkal Woube, Zakaria Hassan, Temesgen Samuel, Abiodun Adesiyun, Charlene R. Jackson, Woubit Abebe

**Affiliations:** 1Center for Food Animal Health, Food Safety and Defense, Department of Pathobiology, College of Veterinary Medicine, Tuskegee University, Tuskegee, AL 36088, USA; rfraj@tuskegee.edu (R.F.); kbentum8786@tuskegee.edu (K.E.B.); bilalf4@yahoo.com (B.A.); ywoube@tuskegee.edu (Y.W.); zico76us@yahoo.com (Z.H.); tsamuel@tuskegee.edu (T.S.); 2Hygiene and Zoonoses Department, Faculty of Veterinary Medicine, Mansoura University, Mansoura 35516, Egypt; hazem_hassan@mans.edu.eg; 3Department of Food Hygiene and Control, Faculty of Veterinary Medicine, University of Sadat City, Sadat City 32511, Egypt; 4Faculty of Medical Sciences, School of Veterinary Medicine, University of the West Indies, St. Augustine 999183, Trinidad and Tobago; abiodun.adesiyun@sta.uwi.edu; 5Poultry Microbiological Safety and Processing Research Unit USDA-ARS, U.S. National Poultry Research Center, Athens, GA 30605, USA; charlene.jackson@usda.gov

**Keywords:** food poisoning, Staphylococcal Enterotoxin (SE), antibiotic resistance, spa typing

## Abstract

Antibiotic-resistant *Staphylococcus aureus* (*S. aureus*) in retail meat poses a public health threat requiring continuous surveillance. This study investigated the frequency of isolation, toxin genes, and antibiotic resistance profile of *S. aureus* recovered from retail poultry meat samples and presented results beneficial to public health interventions. Of 200 samples collected, 16% (32/200) tested positive for *S. aureus*, and these were recovered from thigh 37.5% (12/32), wing 34.4% (11/32), gizzard (15.6% (5/32), and liver 12.5% (4/32) samples. Findings of spa typing analysis revealed that 68.8% (22/32), 18.8% (6/32), 9.4% (3/32), and 3.0% (1/32) of the isolates belonged to the spa types t267, t160, t548, and t008, respectively. For antibiotic susceptibility testing, 12.5% (4/32) of the isolates were resistant to only penicillin, but one isolate (1/32; 3%) showed resistance to the antibiotics penicillin, erythromycin, ampicillin, and oxacillin. PCR analysis revealed that 9.4% (3/32) of the isolates carried the mecA gene associated with methicillin-resistant *Staphylococcus aureus* (MRSA) isolates. One MRSA isolate was identified as a t008 spa type, and harbored a 26,974 bp-sized plasmid, which was the source of its resistance to penicillin, ampicillin, erythromycin, and oxacillin. The staphylococcal enterotoxin (SE) genes *seg*, *sei*, *sek*, *seb*, *selm*, and *seln* were also identified among the isolates, and mostly the antimicrobial and enterotoxin genes were carried on plasmids of the isolates. This study raises awareness on the continuous circulation of pathogenic microbes like *S. aureus* in retail poultry meat.

## 1. Introduction

The U.S. is the largest producer and the second-largest exporter of poultry meat in the world [1]. This makes the poultry industry a significant contributor to the country’s economy. Between 2013 and 2022 alone, approximately 17% of domestic broiler production was exported [2]. For a long time, broiler operations in the U.S. have been concentrated in the country’s southern parts [3], with Georgia, Alabama, Arkansas, North Carolina, and Mississippi still among the leading states, even in recent times [4].

Chicken meat is a product with a higher per capita consumption in the U.S. than beef and pork [5]. However, the increase in the demand for poultry consumption has been associated with an increase in the use of antimicrobials during production [6]. Since 1950, the poultry industry in the U.S. has used antibiotic growth promoters at sub-therapeutic levels as feed additives, to improve performance in terms of feed conversion and weight gain. Despite existing restrictions, antibiotics such as tetracyclines, β-lactams, macrolides, and lincosamides continue to be permitted for use in sub-therapeutic doses to promote animal weight gain [7,8]. Such sub-therapeutic usage of antibiotics has been a breeding ground for antibiotic resistance [9], one of the biggest threats to global health and food security [10]. Currently, over a hundred antimicrobials including beta-lactams, aminoglycosides, tetracyclines, amphenicols, macrolides, sulfonamides, fluoroquinolones, lincosamides, polypeptides, and polyene, have been used globally in food-producing animals [11].

This is alarming, considering the rate at which food pathogens have developed resistance against some of these antibiotics in recent years [12]. *Staphylococcus aureus* (*S. aureus*), with methicillin-resistant *S. aureus* (MRSA) featuring prominently in epidemics [13], is one of such notorious pathogens known for their ability to build resistance. The pathogen is the causative agent of staphylococcal food poisoning (SFP), a common global foodborne illness [14]. *S. aureus* is normally considered a frequent contaminant of retail chicken due to poor hygienic handling of chicken meat [15]. Countries may have different acceptable limits for *S. aureus* counts in chicken meat [16]. However, because SFP is caused by the ingestion of toxins of *S. aureus*, the acceptable level of *S. aureus*, particularly in ready-to-eat foods, should be below 10^3^ colony-forming units per gram (cfu/g) of food [17].

The control of SFP is a challenge, as the continuous change in food habits and processing contributes to the evolution of new strains, exhibiting increased virulence and resistance to applicable preservation methods [18]. Furthermore, the extreme malleability of their genomes and vast potential for adaptability have resulted in increased amounts of antimicrobial-resistant strains of *S. aureus* in all sectors, including those found in food [19]. Poultry has been frequently implicated as a source of SFP [20]. This is partly because the commodity provides a supportive growth environment for *S. aureus* [21].

The pathogenicity of *S. aureus* includes the ability of the bacterium to produce a variety of toxins, which is mostly aimed at helping it evade host defenses [22]. So far, more than 20 different Staphylococcal Enterotoxins (SEs) have been described [23]. These toxins belong to a large family of pyrogenic superantigens [24,25] and are responsible for toxic shock syndrome, a severe condition characterized by rashes, hypovolemic shock, and respiratory distress syndrome [26]. SEs resist proteolytic enzymes such as pepsin or trypsin and tolerate low pH, enabling them to be fully active in the gastrointestinal tract after ingestion [20,24,25].

Furthermore, understanding microbial population dynamics is an important step in disease epidemiology. This is much needed when tracing the source and spread of microorganisms. To achieve this, applying a higher discriminatory power methodology, such as bacterial strain typing, has proven quite beneficial [27]. For the characterization of *S. aureus*, molecular approaches such as spa typing have been crucial in determining prevalent strains for rapid control and preventive measures [28,29]. The spa typing technique relies on assessing the number and sequence variation in repeats at the X-region of the spa gene. It has been a helpful typing tool due to its ease of performance, inexpensive procedure, and standardized nomenclature [29,30,31].

There is a growing demand for chicken meat consumption, coinciding with a rise in antibiotic resistance. This demands that the microbial safety of chicken meat be paralleled with this steady increase in chicken meat consumption [31]. Therefore, this study aimed to determine the frequency and diversity of *S. aureus* isolated from chicken meat samples using spa typing, and to investigate the presence of virulence factors (enterotoxins and antimicrobial resistance genes) in identified isolates.

## 2. Materials and Methods

### 2.1. Samples Collection

A total of 200 fresh chicken meat samples, including liver (n = 48), gizzard (n = 50), thigh (n = 52), and wings (n = 50), were purchased randomly from multiple grocery stores at four different sites in Alabama, USA, from July to October 2019. All collected samples were placed in sterile plastic bags, tightly sealed, and transported in a cold box to the laboratory at the Center for Food Safety and Molecular Biology Lab, Department of Pathobiology, College of Veterinary Medicine, Tuskegee University. Samples were then subjected to microbiological isolation analysis within 24 h after collection.

### 2.2. Staphylococcus aureus Isolation and Identification

To obtain a 1:10 dilution ration for sample enrichment, ten grams of chicken meat for each sample was mixed with 100 mL of buffered peptone water (Neogen, Lansing, MI, USA) and homogenized [32]. Sample homogenization was performed using a Stomacher 400 circulator (Seward Laboratory System, Bohemia, NY, USA) at 250 rpm for 1 min and incubated overnight at 37 °C. After enrichment, a loopful was streaked onto Baird Parker Agar (Hardy Diagnostics, Santa Maria, CA, USA) and incubated overnight at 37 °C. Two to three suspected Staphylococcus colonies, with black morphologies surrounded by 2 to 5 mm clear zones, were selected and subcultured onto Mannitol Salt Agar plates (Difco^®^, Detroit, MI, USA), for further confirmation of *S. aureus.* Individual colonies of presumptive *S. aureus* were further examined using Gram staining and biochemical tests, such as sugar fermentation, indole, coagulase, methyl red, Voges–Proskauer, and the DNase test [33]. ANOVA statistics were performed on the results obtained using GraphPad Prism version 10.3, and a *p*-value < 0.05 was deemed significant.

### 2.3. Bacterial Strains and Oligonucleotide Primers

*S. aureus* subsp. *aureus* Rosenbach (ATCC^®^BAA-1720-MRSA252) was purchased from ATCC (ATCC, Manassas, VA, USA) and used as a reference strain. Primers for analysis of staphylococcal enterotoxins (SEA, SEB, SEC, SED, SEE, SEG, SEH, SEK, SEI, SEJ, SEL, and SER), the staphylococcal-like toxins (SElM, SElN, SElJ, and SElU), and antimicrobial resistance genes (*tetA*, *tetM*, *ermA*, *mecA*, *norA*, *blaZ*, and *chlA*) were adopted from previous studies [34,35,36,37,38,39,40,41,42,43,44,45] (Table 1 and Table 2). The results were visualized using the gplots package (version 3.1.3.1) in R software (version 4.4.1) [45,46] and Microsoft Excel [47].

### 2.4. Genomic and Plasmid DNA Extraction

All positive samples of *S. aureus* and the reference *S. aureus* subsp. *aureus* Rosenbach were grown in tryptic soy broth (Difco^®^, Detroit, MI, USA), placed on a shaker incubator under aerobic conditions at 37 °C/200 rpm overnight, and allowed to grow to mid-log phase. For genomic DNA extraction, 2 mL of bacterial broth culture was centrifuged for 5 min at 10,000 rpm. The supernatant was carefully discarded, and the pellet was used for DNA extraction using the DNeasy^®^ kit, following the manufacturer’s instructions (Qiagen, Valencia, CA, USA). For plasmid DNA extraction, 5 mL of bacterial culture was collected in a 15 mL centrifuge tube and spun for 10 min at 4000 rpm. The supernatant was carefully discarded, and the pellet was used for plasmid DNA extraction using a Qiagen Miniprep kit, following the procedure recommended by the manufacturer (Qiagen, Valencia, CA, USA). The quality and quantity of both genomic and plasmid DNA were determined using a Nanodrop 2000c Spectrophotometer (Thermo Fisher Scientific Inc., Worcester, MA, USA). All extracted DNA was stored at −20 °C for further analysis

### 2.5. Antibiotic Susceptibility Assay

All *S. aureus* isolates and the reference *S. aureus* subsp. *aureus* Rosenbach were tested for antibiotic susceptibility in Mueller Hinton broth, according to the manufacturer’s procedure for minimum inhibitory concentration (MIC) against 18 antimicrobial agents. The MIC was interpreted using the Thermo Scientific™ Sensititre™ semiautomated antimicrobial susceptibility system and the GPN3F Sensititre Gram-positive plate, according to the manufacturer’s directions (Thermofisher Scientific, Carlsbad, CA, USA). The antimicrobial agents included ampicillin, ceftriaxone, ciprofloxacin, clindamycin, daptomycin, erythromycin, gatifloxacin, gentamicin, levofloxacin, linezolid, oxacillin, penicillin, rifampin, tetracycline, vancomycin, trimethoprim/sulfamethoxazole, streptomycin1000, and quinupristin/dalfopristin. The antimicrobial gradient method was performed according to the manufacturer’s recommendation (bioMérieux, Lombard, IL, USA) for the antimicrobial agents norfloxacin and chloramphenicol. The results were interpreted according to the Clinical and Laboratory Standards Institute (CLSI, 2024).

### 2.6. PCR Assay for Enterotoxins and Antibiotic Resistance Genes

All PCR assays were conducted on separate genomic and plasmid DNA of each sample, using a total reaction volume of 20 µL, which consisted of 1 µL (30 ng/µL) of DNA template, 10 µL of the 2X PWO master mix containing DNA polymerase, reaction buffer with 4 mM MgCl2 and 0.4 mM each of PCR-grade dNTPs (Roche, Mannheim, Germany), 7 µL of the PCR grade water, and 1 µL (10 µM) each of forward and reverse primers. PCR amplification was performed with thermocycling conditions of initial denaturation at 95 °C for 5 min, followed by 30 cycles each of denaturation at 95 °C for 1 min, annealing at 58 °C for 1 min, extension at 72 °C for 1 min, followed by a final extension at 72 °C for 10 min. Expected fragment sizes for each of the enterotoxin and antimicrobial resistance genes were visualized using Alpha-Imager (Alpha Innotech Corporation, San Leandro, CA, USA) after fragments were electrophoresed in 1.5% agarose stained with Gel-Red (Biotium, Inc. Fremont, CA, USA). DNA from the reference strain was used as a positive control, and expected band sizes for all gel electrophoresis were measured against a 100 bp ladder.

### 2.7. DNA Preparation and PCR Amplification of Spa Gene

A loopful of *S. aureus* pure colonies was washed with distilled water and incubated with 200 µL of 6% InstaGeneTM Matrix solution (BIO-RAD, München, Germany) for 20 min at 56 °C. The suspensions were vortexed and heated for 8 min at 100 °C and centrifuged at 8000× *g* for 2–3 min. 20 µL of the supernatant containing the DNA was used for PCR amplification for each sample. DNA sequencing of the spa genes for each isolate was carried out at the US National Poultry Research Center, Athens, GA, USA, and spa typing was performed as previously described [50,51].

### 2.8. Sequencing and Analysis of the Spa Type t008 Isolate’s Plasmid

The extracted plasmid DNA was sequenced using a MinION Mk1B device (Oxford Nanopore Technologies, Oxford, UK). Following the manufacturer’s protocol, the plasmid DNA library was prepared using the rapid sequencing kit (SQK-RAD004). The generated libraries in a sequencing buffer were loaded into a primed MinION R9.4.1 flow cell (active pores number > 800) via the sample port and sequenced for 4 h. Base calling on the sequenced reads was performed using Guppy (v6.5.7) incorporated in the MinKNOW software v3.1.20. The quality of the reads was assessed using FastQC (version 0.11.3) [52], trimmed using Porechop v0.2.3 [53], and assembled using Flye software v2.9.1 [54], with iterations and threads set at 2 and 10, respectively. The circular plasmid contig generated from the assembly was confirmed using NCBI Blast (https://blast.ncbi.nlm.nih.gov/Blast.cgi). Finally, resistance genes on the identified plasmid were determined using AMRfinder_v3.11.20, with database version 2023-11-15.1 (Accessed on 16 November 2024) [55].

## 3. Results

### 3.1. Frequency of S. aureus Isolation in Chicken Meat Samples

Of the two hundred samples collected, 16% (32/200) were positive for *S. aureus,* with all four sample types yielding at least one positive isolate (Appendix A). The frequency of *S. aureus* isolation in the tested chicken meat products was highest in chicken thigh samples at 37.5% (12/32), followed by chicken wings at 34.4% (11/32), gizzard at 15.6% (5/32), and liver at 12.5% (4/32) (Figure 1).

### 3.2. Minimum Inhibitory Concentration Assay and Antimicrobial Resistance Genes

Most isolates (96.9% (31/32)) carried at least one of the antimicrobial-resistant genes we tested. The single isolate negative for all the tested genes was recovered from a chicken thigh sample (Figure 2). In total, 3.1% (1/32), 93.8% (30/32), 43.8% (14/32), 84.4% (27/32), 9.4% (3/32), and 81.3% (26/32) of the isolates carried the resistant genes *tetM*, *tetA*, *blaZ*, *norA*, *mecA*, *ermA*, and *chlA*, respectively. The three isolates identified as MRSA strains for carrying the *mecA* gene [56] were each recovered from a chicken wing, thigh, and gizzard sample, respectively (Figure 2).

For antimicrobial susceptibility, resistance was detected in five of the isolates against four (penicillin, ampicillin, erythromycin, and oxacillin) of the antimicrobials tested. Two of these resistant isolates were from chicken thigh, two from chicken liver, and one from chicken wing samples. All five isolates showed resistance to penicillin. However, one identified MRSA isolate from a chicken wing sample was, in addition to penicillin, resistant to ampicillin, erythromycin, and oxacillin. This observation was unsurprising, as isolates resistant to oxacillin are usually resistant to other β-lactams (Figure 3).

### 3.3. Distribution of Enterotoxin Genes Within the Plasmids of S. aureus Isolates

The enterotoxin genes *seg*, *sei*, *selm*, *seln*, *seb*, and *sek* were identified in four isolates. Two of these isolates were from chicken wing samples, each harboring the *seg*, *sei*, *selm*, and *seln* encoding genes concurrently. The *sek* and *seb* genes were each carried separately by two isolates recovered from chicken wing and chicken liver samples, respectively (Figure 4). Interestingly, the enterotoxin genes detected in the current study were all carried on plasmids from isolated organisms.

### 3.4. Multidrug Resistance Plasmid Carried Genes Against Three Antibiotic Classes

Plasmid DNA from one of the isolates (spa type t0080) was sequenced. After genome assembly, two circular contigs of 26,974 bp and 3120 bp were identified as *S. aureus* plasmids. The antimicrobial resistance genes *ant(6)-Ia*, *aph(3′)-III*, *blaZ*, *mph(C)*, and *msr(A)*, conferring resistance to aminoglycosides, beta-lactams, and macrolides, were identified on the larger of the two plasmid contigs (Appendix A). All raw sequence data have been deposited at the NCBI under SRR accession SRR29790551 and BioProject accession number PRJNA1134810 (Reviewer link https://dataview.ncbi.nlm.nih.gov/object/PRJNA1134810?reviewer=9a5uc982869i7u4a69230nth8f (Accessed 11 July 2024)).

### 3.5. Molecular Typing of Spa Gene

Analysis of the spa gene showed that the 32 *S. aureus* isolates each belonged to one of four spa types: t267, t160, t548, and t008. The predominant spa type to which 68.8% (22/32) of the isolates belonged was t267. This was followed by t160, t548, and t008 in the proportions 18.8% (6/32), 9.4% (3/32), and 3.1% (1/32), respectively (Figure 5).

## 4. Discussion

The frequency of *S. aureus* isolation from retail chicken meat samples determined in this study was 16% (32/200). This result is comparable to another study in the U.S., which reported that 17.8% of retail chicken meat sampled was positive for *S. aureus* [57]. Nevertheless, others have reported an even higher percentage of about 41% in meat and poultry samples in the U.S. [58]. The difference in the results may be influenced by factors such as sampling size and the location and methodology of the study. Although *S. aureus* is normally present on birds’ skin and in their internal organs [59], an observation of this study was that it was more common to isolate *S. aureus* from chicken thigh and wing samples compared to gizzard and liver samples, which are internal organs of birds. This is possibly because carcasses and cuts during poultry processing can be contaminated by the resident microbes, such as *Campylobacter*, *Salmonella*, and *Listeria monocytogenes*, commonly found in the slaughterhouse environment, and this can be from equipment surfaces, the water used, and even bacteria from the air [31]. There is a significant risk of human contamination from these external sources during meat handling or processing [60].

Staphylococcal Enterotoxins (SEs) are exotoxins produced by *S. aureus*, and are responsible for SFP in humans worldwide. Toxins that induce a vomitory effect are sometimes referred to as SEs. In contrast, those that cannot induce a vomitory effect on primate models have been referred to as staphylococcal-like enterotoxins (SEls) [23]. This study identified four SE genes (*seb*, *sek*, *seg*, and *sei*) and two SEl genes (*selm* and *seln*). Some researchers have explored the emetic potential of the toxins SElM and SElN in monkeys, concluding that they may play some kind of role in SFP [61]. The emetic potential of SEs is very important in causing SFP, aside from them being considered super-antigens [61,62].

The presence of resistant bacteria in chickens poses a significant risk to human health. Four out of five *S. aureus* isolates identified in this study were resistant only to penicillin, with the remaining isolate (a MRSA strain) resistant to penicillin, ampicillin, erythromycin, and oxacillin. As mentioned, such a demonstration of *S. aureus* resistance against beta-lactam agents has also been reported in other studies [58]. Although *S. aureus* isolated in this study possessed the resistant genes *tetM*, *tetA*, *blaZ*, *norA*, *mecA*, and *chlA*, antimicrobial resistance was observed against only penicillin, ampicillin, erythromycin, and oxacillin. We infer that these genes, while present, were not phenotypically expressed. This could be a phenomenon of “cryptic genes” [63], where genes, although present, are silent and unexpressed. Although the *ermA* gene was not detected in the isolates, the observed erythromycin resistance can be attributed to other genes or resistance mechanisms not investigated in this study. The *ermB*, *ermC*, and *msrA* genes, for instance, have also been implicated in erythromycin resistance [64].

A noteworthy finding was that almost all the antimicrobial genes (except mecA), including the plasmid-associated toxin and enterotoxin virulence genes, were carried on the plasmids of the isolates. For example, one MRSA isolate carried five resistance genes (*ant(6)-Ia*, *aph(3′)-III*, *blaZ*, *mph(C)*, and *msr(A)*) on a single plasmid of 26,974 bp, which provides resistance to aminoglycosides, beta-lactams, and macrolides. Plasmids are recognized as mobile genetic elements in *S. aureus*, enabling it to rapidly adapt to the selective pressures imposed by humans [65]. Different studies have demonstrated that plasmids carry diverse levels of antimicrobial resistance and can also carry toxin genes [66,67,68]. The finding of plasmid-associated toxin and antimicrobial resistance genes aligns with the study by McCarthy and Lindsay, who sequenced and analyzed 253 *S. aureus* plasmids [68].

From spa typing analysis, identifying t008 (MRSA type USA300) from a retail chicken wing sample was significant, as it was also a MRSA isolate. The MRSA population in the U.S. comprises two dominant lineages, USA300 and USA100, each consisting of closely related variants [28]. The USA300 (ST8/t008) has been a common cause of *S. aureus* infection in children, and is also commonly associated with skin and soft tissues and outpatient environments [69]. Furthermore, t008/USA300 isolates have also been found in meat samples in the Netherlands [70,71], but to the best of our knowledge, this study presents the first report on the isolation of this spa type in retail poultry in Alabama. The other dominant spa types, t160 and t267, have been isolated from humans in previous studies [72], and in some cases from milk samples associated with mastitis [35]. The spa type t548 has been isolated from chicken meat samples in other research [40], with its MRSA variant also isolated from ready-to-eat fish products and humans [73,74]. Our findings are well supported by these other studies, which have also reported the circulation of various spa types of *S. aureus* and MRSA strains in food samples and humans.

It is not uncommon to isolate *S. aureus* from retail chicken samples, and the acceptable limits of *S. aureus* counts in chicken meat have been determined. Nevertheless, the continuous circulation of MRSA strains is quite worrying, as the pathogen is considered a public health threat globally [75]. Therefore, this is a wake-up call to consumers and stakeholders to ensure hygienic standards are well adhered to when handling retail chicken meat.

## 5. Conclusions

In conclusion, this study indicates that MRSA and other virulent and antimicrobial-resistant strains of *S. aureus* can potentially enter and spread in the food chain through retail chicken meat. It is therefore important for consumers to practice safe handling methods and cook retail meat well.

## Figures and Tables

**Figure 1 pathogens-14-00107-f001:**
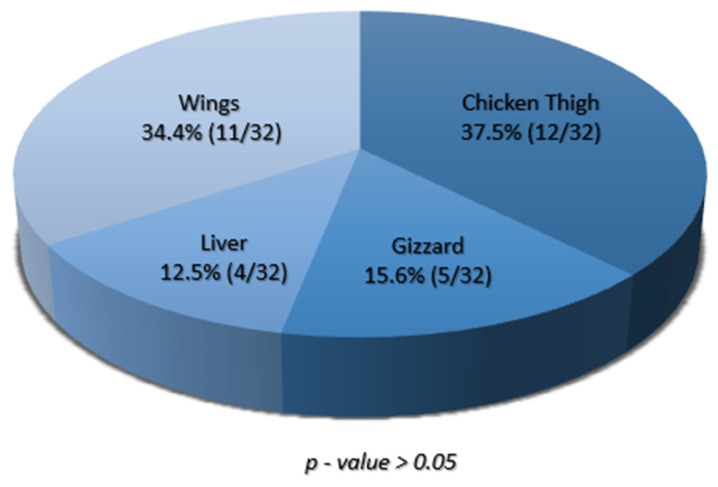
A pie chart showing the proportions of *S. aureus* isolated from various chicken meat parts. Each section of the chart shows the percentage of isolates recovered from the given chicken part, and the corresponding proportions are shown in brackets.

**Figure 2 pathogens-14-00107-f002:**
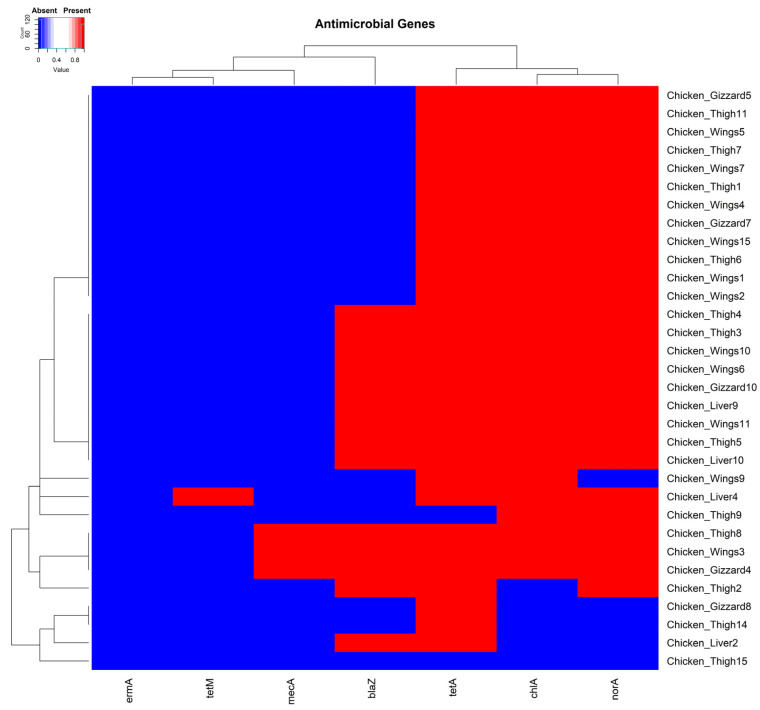
A cluster dendrogram with a heatmap showing the presence or absence of antimicrobial genes in various samples. Sample identifiers are shown to the right and antimicrobial genes (*tetA*, *tetM*, *ermA*, *mecA*, *norA*, *blaZ*, and *chlA*) are shown at the bottom of the diagram. Red and blue depict the presence and absence, respectively, of the corresponding genes in the isolates of the samples.

**Figure 3 pathogens-14-00107-f003:**
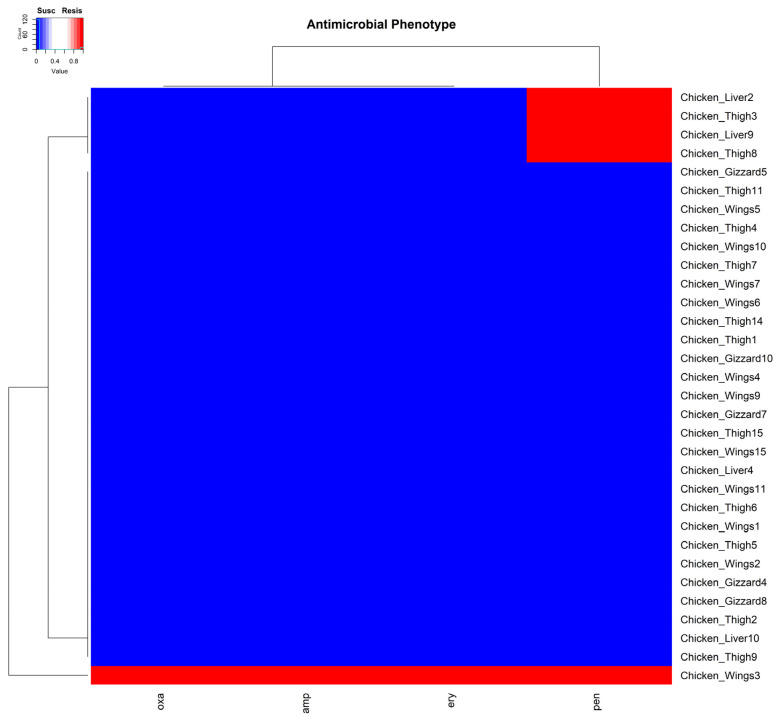
A cluster dendrogram with a heatmap showing the resistance or susceptibility of samples to antibiotics. Sample identifiers are shown to the right and the antibiotics (penicillin (pen), ampicillin (amp), erythromycin (ery), and oxacillin (oxa) are shown at the bottom of the diagram. Red and blue depict resistance (Resis) and susceptibility (Susc) to the corresponding antibiotic by the isolate from the samples.

**Figure 4 pathogens-14-00107-f004:**
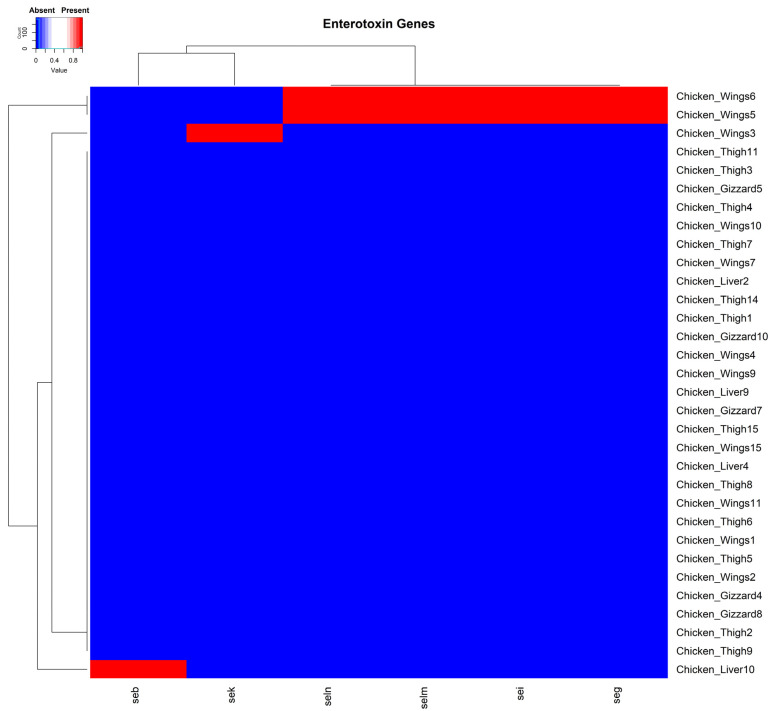
A cluster dendrogram with a heatmap showing the presence or absence of staphylococcal enterotoxin genes in samples. Sample identifiers are shown to the right, and the enterotoxin genes *seg*, *sei*, *sek*, *seb*, *selm*, and *seln* are shown at the bottom of the diagram. Red and blue depict the presence and absence, respectively, of the corresponding genes in the isolates of the various samples.

**Figure 5 pathogens-14-00107-f005:**
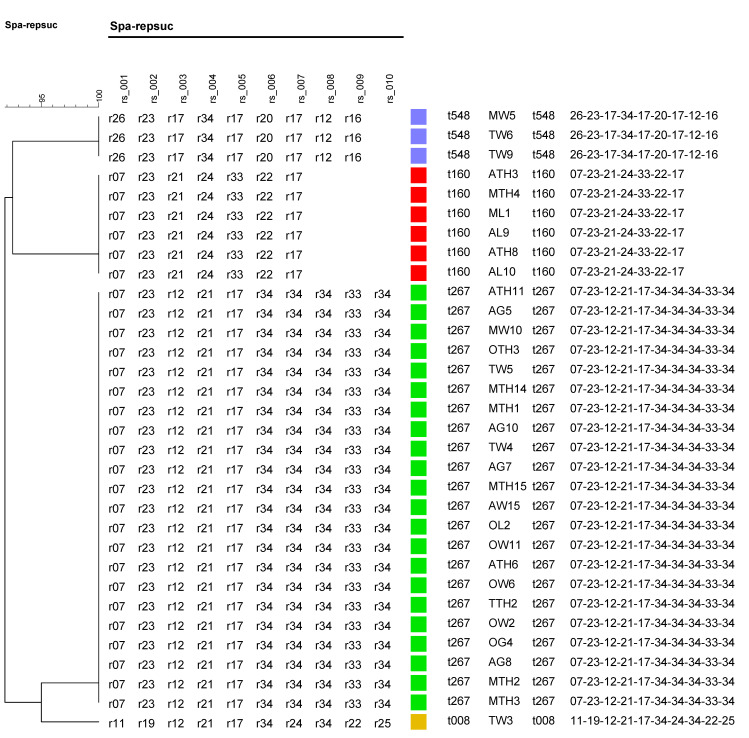
Dendrogram tree based on spa typing results of isolates from samples. The various spa types and their associated samples are shown to the right. The spa types t548, t160, t267, and t008 are also represented by the colors violet, red, green, and yellow.

**Table 1 pathogens-14-00107-t001:** Oligonucleotide sequences used for enterotoxin gene analysis.

Enterotoxins	Oligonucleotide Sequences (5′-3′)	Expected Size bp	Reference
	F and R Primers		
SEA	CCTTTGGAAACGGTTAAAACG	127	[41]
	TCTGAACCTTCCCATCAAAAAC		
SEB	TCGCATCAAACTGACAAACGA	410	[48]
	CACTTTTTCTTTGTCGTAAGATAA		
SEC	AGATGAAGTAGTTGATGTGTATGG	451	[48]
	CACACTTTTAGAATCAACCG		
SED	CCAATAATAGGAGAAAATAAAAG	278	[48]
	ATTGGTATTTTTTTTCGTTC		
SEE	CAGTACCTATAGATAAAGTTAAAACAAGC	178	[48]
	TAACTTACCGTGGACCCTTC		
SEG	GCTATCGACACACTACAACC′	538	[48]
	CCAAGTGATTGTCTATTGTCG		
SEH	CACATCATATGCGAAAGC	584	[48]
	CGAATGAGTAATCTCTAGG		
SEI	GATACTGGAACAGGACAAGC	789	[34]
	CTTACAGGCAGTCCATCTCC		
SEJ	CTCCCTGACGTTAACACTACTAATAA	666	[48]
	TTGTCTGGATATTGACCTATAACATT		
SEK	CACAGCTACTAACGAATATC	378	[48]
	TGGAATTTCTCAGACTCTAC		
SEL	CATACAGTCTTATCTAACGG	275	[48]
	TTTTCTGCTTTAGTAACACC		
SER	AGATGTGTTTGGAATACCCTAT’	123	[36]
	CTATCAGCTGTGGAGTGCAT		
SElM	CTTGTCCTGTTCCAGTATC	329	[48]
	ATACGGTGGAGTTACATTAG		
SElN	GATGAAGAGAAAGTTATAGGCGT	268	[36]
	AACTCTGCTCCCACTGAACC		
SELlU	AATGGCTCTAAAATTGATGGTTC	142	[36]
	CCATATTATCCGCTGAAAAATAG		
SElJ	GTTCTGGTGGTAAACCA	132	[36]
	GCGGAA CAACAGTTCTGA		

**Table 2 pathogens-14-00107-t002:** Oligonucleotide sequences used for antimicrobial resistance gene analysis.

Antibiotic Resistance Genes	Oligonucleotide Sequences (5′-3′)	Expected Size bp	References
	F and R Primers		
*blaZ*	ACTTCAACACCTGCTGCTTTC	173	[44]
	TGACCACTTTTATCAGCAACC		
*mecA*	AAAATCGATGGTAAAGGTTGGC	533	[43]
	AGTTCTGCAGTACCGGATTTGC		
*tetA*	CTAAAGAGGACCAGAAGACT	512	[42]
	ATAACCCGGTACTAATAACA		
*tetM*	ACAGAAAGCTTATTATATAAC	171	[49]
	TGGCGTGTCTATGATGTTCAC		
*chlA*	CCTGCTAACAATAGACCTGA	768	[42]
	CGCTTTAACATTTGCGATAT		
*norA*	TGTTAAGTCTTGGTCATCTGCA	761	[37,38]
	CCATAAATCCACCAATCCC		
*ermA*	CTTCGATAGTTTATTAATATTAG	645	[39]
	TTCTAAAAAGCATGTAAAAGAA		

## Data Availability

Supporting Data of plasmid sequence can be found at https://dataview.ncbi.nlm.nih.gov/object/PRJNA1134810?reviewer=9a5uc982869i7u4a69230nth8f (accessed 11 July 2024).

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
