# Peer review of "Antimicrobial Resistance, Virulence Gene Profiling, and Spa Typing of Staphylococcus aureus Isolated from Retail Chicken Meat in Alabama, USA"

_pathogens, 2025, doi:10.3390/pathogens14020107_

Round 1
Reviewer 1 Report
Comments and Suggestions for Authors
1. Line 48: It is recommended to change “An important contributor” to “A significant contributor” to better highlight its prominent role and impact, more accurately conveying its key contribution to the economy.
2. Line 58-60: It is recommended to replace “Currently, restrictions are in place, yet several antibiotics such as tetracyclines, β-lactams, macrolides, and lincosamides are still allowed for use at sub-therapeutic dosages for animal weight gain purpose” with “Despite existing restrictions, antibiotics such as tetracyclines, β-lactams, macrolides, and lincosamides continue to be permitted for use at sub-therapeutic doses to promote animal weight gain.” This revision is more concise and easier to understand.
3. Please ensure that the scientific name Staphylococcus aureus (S. aureus) is italicized throughout the manuscript.
4. Line 90: Replace “In doing so” with “To achieve this” for a more formal tone.
5. line 97-99: Consider simplifying the sentence for clarity: “Given the increasing demand for chicken meat and rising antibiotic resistance, its microbial safety must keep pace with consumption growth”.
6. Line121: The current identification of Staphylococcus aureus relies on traditional culture and biochemical methods, which, while effective, may lack specificity. It is recommended to incorporate PCR-based detection targeting specific genes such as nuc, spa, or 16S rRNA to improve the accuracy and reliability of the results.
7. Line 171: The use of the CLSI 2018 standard for interpreting MIC results may be outdated. It is recommended to refer to the most recent version of the CLSI guidelines, to ensure more accurate and up-to-date interpretations of the results.
8. Line 192: It is advisable to use the standard abbreviation “20 µl” rather than “Twenty microliters” to ensure consistency throughout the manuscript.
9. Line 220: It would be helpful to add a header to Supplementary Table 1 for clarity. Additionally, for consistency, the column labels for “ % Sample Part Positive ” and “ % Sample Part Negative ”could be revised to “ Sample Part Positive (%) ” and “ Sample Part Negative (%) ”, respectively.
10. Line 220: In Supplementary Table 1, only region D summarizes information for all samples, while regions A, B, and C do not. For consistency, it is recommended that either all regions summarize the information for all samples, or none do, to maintain uniformity throughout the table.
11. Line 220-223: Please change “higher” to “highest” for clarity. Additionally, the meaning of “p-value > 0.05” should be explicitly stated, indicating that the differences between groups are not statistically significant.
12. Line 225-239: According to the methods, the corresponding results should be presented in sequence. It is recommended to first present the antimicrobial resistance data, followed by the discussion of the antimicrobial resistance genes. Additionally, the MIC breakpoints and results for each antibiotic tested should be presented in a table format to better illustrate the findings and enhance the clarity and persuasiveness of the results.
13. Line 289 : The phrase “(selm, and seln)” contains "and" which may be unnecessary.
14. Line 294: The phrase “ resistant to only penicillin” should be revised to “resistant only to penicillin” for better flow and clarity.
15. Line 308-312: The sentence could be reworded for clarity and flow. A suggestion might be: “ For example, one MRSA isolate carried five resistance genes (ant(6)-Ia, aph(3')-III, blaZ, mph(C), and msr(A)) on a single plasmid of 26,974bp, which provides resistance to aminoglycosides, beta-lactams, and macrolides. Plasmids are recognized as mobile genetic elements in S. aureus , enabling it to rapidly adapt to the selective pressures imposed by humans.”
16. line 324: It is recommended to standardize the term to “spa type” throughout the manuscript to avoid inconsistencies between “spa-type” and “spa type, ensuring consistency and professionalism in the text.
Comments on the Quality of English Language
Some parts of the article could be more concise, and certain sections are unclear in their expression, suggesting further revisions to improve clarity and precision.
Author Response
Reviewer #1 1. Line 48: It is recommended to change “An important contributor” to “A significant contributor” to better highlight its prominent role and impact, more accurately conveying its key contribution to the economy. Response: Done 2. Line 58-60: It is recommended to replace “Currently, restrictions are in place, yet several antibiotics such as tetracyclines, β-lactams, macrolides, and lincosamides are still allowed for use at sub-therapeutic dosages for animal weight gain purpose” with “Despite existing restrictions, antibiotics such as tetracyclines, β-lactams, macrolides, and lincosamides continue to be permitted for use at sub-therapeutic doses to promote animal weight gain.” This revision is more concise and easier to understand. Response: Done
3. Please ensure that the scientific name Staphylococcus aureus (S. aureus) is italicized throughout the manuscript. Response: Done
4. Line 90: Replace “In doing so” with “To achieve this” for a more formal tone. Response: Done
5. line 97-99: Consider simplifying the sentence for clarity: “Given the increasing demand for chicken meat and rising antibiotic resistance, its microbial safety must keep pace with consumption growth”. Response: The section has been appropriately revised
6. Line121: The current identification of Staphylococcus aureus relies on traditional culture and biochemical methods, which, while effective, may lack specificity. It is recommended to incorporate PCR-based detection targeting specific genes such as nuc, spa, or 16S rRNA to improve the accuracy and reliability of the results. Response: Thanks for the recommendation. The isolates recovered were confirmed by spa analysis in subsequent analysis as shown in Figure 5.
7. Line 171: The use of the CLSI 2018 standard for interpreting MIC results may be outdated. It is recommended to refer to the most recent version of the CLSI guidelines, to ensure more accurate and up-to-date interpretations of the results. Response: Thanks for the observation. However, this work was conducted between July and October 2019. Hence the CLSI 2018 was used in interpreting our results obtained around the time of the study. However, the results is even true interpreting with CLSI 2024 so we have modified this in the manuscript appropriately.
8. Line 192: It is advisable to use the standard abbreviation “20 µl” rather than “Twenty microliters” to ensure consistency throughout the manuscript. Response: Done
9. Line 220: It would be helpful to add a header to Supplementary Table 1 for clarity. Additionally, for consistency, the column labels for “ % Sample Part Positive ” and “ % Sample Part Negative ”could be revised to “ Sample Part Positive (%) ” and “ Sample Part Negative (%) ”, respectively. Response: Done
10. Line 220: In Supplementary Table 1, only region D summarizes information for all samples, while regions A, B, and C do not. For consistency, it is recommended that either all regions summarize the information for all samples, or none do, to maintain uniformity throughout the table. Response: Thanks for the observation. The summarized information for samples for the other regions have been provided as well.
11. Line 220-223: Please change “higher” to “highest” for clarity. Additionally, the meaning of “p-value > 0.05” should be explicitly stated, indicating that the differences between groups are not statistically significant. Response: Done
12. Line 225-239: According to the methods, the corresponding results should be presented in sequence. It is recommended to first present the antimicrobial resistance data, followed by the discussion of the antimicrobial resistance genes. Additionally, the MIC breakpoints and results for each antibiotic tested should be presented in a table format to better illustrate the findings and enhance the clarity and persuasiveness of the results. Responds: The heading and the section have been reorganized appropriately. Doc please do not know if this can be provided as the original manuscript table only showed Resistant/ sensitive without any values with no MIC data.
13. Line 289: The phrase “(selm, and seln)” contains "and" which may be unnecessary. Response: Done
14. Line 294: The phrase “resistant to only penicillin” should be revised to “resistant only to penicillin” for better flow and clarity. Response: Done
15. Line 308-312: The sentence could be reworded for clarity and flow. A suggestion might be: “For example, one MRSA isolate carried five resistance genes (ant(6)-Ia, aph(3')-III, blaZ, mph(C), and msr(A)) on a single plasmid of 26,974bp, which provides resistance to aminoglycosides, beta-lactams, and macrolides. Plasmids are recognized as mobile genetic elements in S. aureus, enabling it to rapidly adapt to the selective pressures imposed by humans.” Response: The reviewer’s suggestion has been adopted
16. line 324: It is recommended to standardize the term to “spa type” throughout the manuscript to avoid inconsistencies between “spa-type” and “spa type, ensuring consistency and professionalism in the text. Response: Done |

Reviewer 2 Report
Comments and Suggestions for Authors
The manuscript entitled; “Prevalence, antimicrobial resistance, and virulence gene profiles of Staphylococcus aureus isolated from retail chicken meat” is a well-structured and addresses an important topic in S. aureus AMR, food hygiene, public health
please see my comments below.
Title:
The title is clear and accurately reflects the content. To enhance specificity, please mention the location at th end of the title : “…….in Alabama, USA”
Introduction:
The aim is clearly stated and directly addresses a critical gap in the literature. However, the objectives could be more specific by detailing how findings will inform clinical or public health interventions.
Methods, such as molecular techniques and susceptibility testing protocols, are clearly described, ensuring replicability.
Major concern:
Sample size calculation.
I have major concerns on the use of ref. 32 to do the calculation.
Since no data is available on MDR S. aureus in the state of Alabama, it should be based on 50% prevalence as standard protocol for unknown prevalence calculation???
You can not take 26% from other state to do the calculation.
Please recalculate the sample size and do the study
Or simply remove this section and remove the term PREVALENCE from title and other part.
Line 223, how the p value was determined, write in methodology please.
Line 280, what are those resident microbes, name please??
The results are presented systematically, with clear tables and figures. Statistical analyses are appropriate for the data, and the findings are logically interpreted. However, there are not enough discussion on the implications or public health signficance of the findings. Please add few more sentenses in this regard.
Please write the conclusion under a separate heading
The Figures are clear and effectively complement the text. However, Figure legends could provide more detail, especially for molecular results.
Other comments:
Please add plates of sensitivity test and PCR gel images in suppl. File
The manuscript is well-written, with minimal grammatical issues. However, some corrections please as follows:
Line 294, S. aureus make Italic
Line 302, emr italic pls.
Author Response
Title: The title is clear and accurately reflects the content. To enhance specificity, please mention the location at the end of the title: “…….in Alabama, USA” Response: Done
Introduction: The aim is clearly stated and directly addresses a critical gap in the literature. However, the objectives could be more specific by detailing how findings will inform clinical or public health interventions. Response: This has been clarified
Methods, such as molecular techniques and susceptibility testing protocols, are clearly described, ensuring replicability.
Major concern: Sample size calculation. I have major concerns on the use of ref. 32 to do the calculation. Since no data is available on MDR S. aureus in the state of Alabama, it should be based on 50% prevalence as standard protocol for unknown prevalence calculation??? You cannot take 26% from other state to do the calculation. Please recalculate the sample size and do the study Or simply remove this section and remove the term PREVALENCE from title and other part. Response: The suggestion by the reviewer is deem appropriate and has therefore been adopted
Line 223, how the p-value was determined, write in methodology, please. Response: This was obtained by performing ANOVA statistics on the data using Graphpad Prism version 10.3. and it has been included in the methodology
Line 280, what are those resident microbes, name please? Response: Some of them have been named as requested.
The results are presented systematically, with clear tables and figures. Statistical analyses are appropriate for the data, and the findings are logically interpreted. However, there are not enough discussion on the implications or public health significance of the findings. Please add few more sentences in this regard. Response: Thanks for the suggestions we consider the current discussion to suffice for the information we wish to communicate.
Please write the conclusion under a separate heading Response: This has been done
The Figures are clear and effectively complement the text. However, Figure legends could provide more detail, especially for molecular results. Response: Thanks for the comments. Molecular results have been provided in the supplementary data.
Other comments: Please add plates of sensitivity test and PCR gel images in suppl. File Response: Thanks for pointing that out. Gel images have been supplied as part of the supplementary data and the MIC results obtained from the sensitivity analysis as well
The manuscript is well-written, with minimal grammatical issues. However, some corrections please as follows:
Line 294, S. aureus make Italic Response: Done
Line 302, emr italic pls. Response: Done |

Reviewer 3 Report
Comments and Suggestions for Authors
The article entitled "Prevalence, antimicrobial resistance, and virulence gene profiles of Staphylococcus aureus isolated from retail chicken meat" submitted for consideration to Pathogens presents the results of a study on the prevalence of S. aureus in chicken carcasses. Although this is a topic that has already been widely evaluated by the global scientific community, the authors explore the capacity of isolates to be resistant to antibiotics and to possess resistance and virulence genes. The study presents some inconsistencies, especially when it highlights the high frequency of S. aureus, which I do not agree with, since what is recommended for food is to count this pathogen, as there are microbiological limits for this. Therefore, I am sending an attached document with my considerations. In order to be accepted, the study needs to be restructured so as not to give the impression that isolating S. aureus is an alarming situation.

Author Response
Line 1: Response: “Title” has been removed, the caption modified to reflect origin of samples and all Staphylococcus aureus have been italicized throughout the manuscript
Line 23: Please provide more details about the materials and methods. Number of samples, collection period, analyses performed, data processing, etc. Response: To stay within the word limit for the abstract, the requested details have been provided in the methodology.
Linie 25: Is positivity alone important for this type of food? Wouldn't it be more important to determine the quantity since it is expected that processed foods will have a certain level of S. aureus contamination. Response: While we recognize the importance of determining the quantity, it was beyond the scope of our study due to resource limitations. Our objective was solely to assess the frequency of pathogen isolation from the samples.
Line 28: Which and how many antibiotics were tested? Response: Again, to stay within the word limit for the abstract, the requested details have been provided in the methodology.
Line 69: What is the microbiological standard for S. aureus in this type of food? Cite methodology. Response: Thanks for the comment. The manuscript has been revised to provide this information, and this has been appropriately cited
Line 139: Please indicate statistical analyses in a separate item at the end of all analyses performed Response: All statistical analysis were performed using ANOVA in GraphPad Prism version 10.3. and it has been included in the methodology
Line 218: ... or frequency? Response: We agree with the reviewer’s suggestion and have adopted it appropriately as our estimation of the sample size coupled with limited resources will not allow us to perform an efficient prevalence analysis |

Reviewer 4 Report
Comments and Suggestions for Authors
This study addresses a significant public health concern regarding the prevalence and antimicrobial resistance of Staphylococcus aureus in retail chicken meat. The investigation is timely, given the increasing rates of antibiotic resistance globally. The authors have employed appropriate methodologies, including spa typing and PCR analysis, to assess the presence of resistant strains and virulence factors. Overall, the article is well-structured and presents valuable findings.
Grammatical and Stylistic Issues
Abstract: "the presence of antibiotic-resistant Staphylococcus aureus (S. aureus) in retail meat is a public health threat requiring continuous surveillance." Suggestion: "the presence of antibiotic-resistant Staphylococcus aureus (S. aureus) in retail meat poses a public health threat that requires continuous surveillance."
Introduction: "The U.S. is the largest producer and the second-largest exporter of poultry meat globally." Suggestion: "The U.S. is the largest producer and the second-largest exporter of poultry meat in the world."
Methodology: "Due to the lack of available research data on multidrug-resistant S. aureus in the state of Alabama, an assumed prevalence (Pexp) of 26% from chicken (breast & thighs) taken from a previous study was used to determine the sample size for this study." Suggestion: "Due to the lack of available research data on multidrug-resistant *S. aureus* in Alabama, an assumed prevalence (Pexp) of 26% from a previous study on chicken (breast and thighs) was used to determine the sample size.
Results: "The prevalence of S. aureus in the tested chicken meat products was higher in chicken thigh samples, 37.5% (12/32), followed by chicken wings at 34.4% (11/32)..." Suggestion: "The prevalence of S. aureus in the tested chicken meat products was highest in chicken thigh samples at 37.5% (12/32), followed by chicken wings at 34.4% (11/32)..."
Section 3.4: "The plasmid DNA from one of the isolates (spa-type t0080) was sequenced." Suggestion: "Plasmid DNA from one of the isolates (spa-type t0080) was sequenced."
Discussion: "This result is comparatively similar to another study in the U.S. which reported an S. aureus prevalence of 17.8% in retail chicken." Suggestion: "This result is comparable to another study in the U.S. that reported an S. aureus prevalence of 17.8% in retail chicken."
"The toxins with the ability to induce a vomitory effect have sometimes specifically been referred to as SE..." Suggestion: "Toxins that induce a vomitory effect are sometimes referred to as SE..."
"The presence of resistant bacteria in chickens poses a significant risk to human health." Suggestion: "The presence of resistant bacteria in chickens poses a significant risk to human health."
"We infer that these genes, although present, were not phenotypically expressed." Suggestion: "We infer that these genes, while present, were not phenotypically expressed."
"A worth-noting finding was that almost all the antimicrobial genes (except mecA)..." Suggestion: "A noteworthy finding was that almost all the antimicrobial genes (except mecA)..."
"The finding of plasmid-associated toxin and antimicrobial resistance genes agrees with the study of McCarthy and Lindsay who sequenced and studied 253 S. aureus plasmids." Suggestion: "The finding of plasmid-associated toxin and antimicrobial resistance genes aligns with the study by McCarthy and Lindsay, who sequenced and analyzed 253 S. aureus plasmids."
Author Response
Comments 1: [Paste the full reviewer comment here.] Reviewer #1 1. Line 48: It is recommended to change “An important contributor” to “A significant contributor” to better highlight its prominent role and impact, more accurately conveying its key contribution to the economy. Response: Done 2. Line 58-60: It is recommended to replace “Currently, restrictions are in place, yet several antibiotics such as tetracyclines, β-lactams, macrolides, and lincosamides are still allowed for use at sub-therapeutic dosages for animal weight gain purpose” with “Despite existing restrictions, antibiotics such as tetracyclines, β-lactams, macrolides, and lincosamides continue to be permitted for use at sub-therapeutic doses to promote animal weight gain.” This revision is more concise and easier to understand. Response: Done
3. Please ensure that the scientific name Staphylococcus aureus (S. aureus) is italicized throughout the manuscript. Response: Done
4. Line 90: Replace “In doing so” with “To achieve this” for a more formal tone. Response: Done
5. line 97-99: Consider simplifying the sentence for clarity: “Given the increasing demand for chicken meat and rising antibiotic resistance, its microbial safety must keep pace with consumption growth”. Response: The section has been appropriately revised
6. Line121: The current identification of Staphylococcus aureus relies on traditional culture and biochemical methods, which, while effective, may lack specificity. It is recommended to incorporate PCR-based detection targeting specific genes such as nuc, spa, or 16S rRNA to improve the accuracy and reliability of the results. Response: Thanks for the recommendation. The isolates recovered were confirmed by spa analysis in subsequent analysis as shown in Figure 5.
7. Line 171: The use of the CLSI 2018 standard for interpreting MIC results may be outdated. It is recommended to refer to the most recent version of the CLSI guidelines, to ensure more accurate and up-to-date interpretations of the results. Response: Thanks for the observation. However, this work was conducted between July and October 2019. Hence the CLSI 2018 was used in interpreting our results obtained around the time of the study. However, the results is even true interpreting with CLSI 2024 so we have modified this in the manuscript appropriately.
8. Line 192: It is advisable to use the standard abbreviation “20 µl” rather than “Twenty microliters” to ensure consistency throughout the manuscript. Response: Done
9. Line 220: It would be helpful to add a header to Supplementary Table 1 for clarity. Additionally, for consistency, the column labels for “ % Sample Part Positive ” and “ % Sample Part Negative ”could be revised to “ Sample Part Positive (%) ” and “ Sample Part Negative (%) ”, respectively. Response: Done
10. Line 220: In Supplementary Table 1, only region D summarizes information for all samples, while regions A, B, and C do not. For consistency, it is recommended that either all regions summarize the information for all samples, or none do, to maintain uniformity throughout the table. Response: Thanks for the observation. The summarized information for samples for the other regions have been provided as well.
11. Line 220-223: Please change “higher” to “highest” for clarity. Additionally, the meaning of “p-value > 0.05” should be explicitly stated, indicating that the differences between groups are not statistically significant. Response: Done
12. Line 225-239: According to the methods, the corresponding results should be presented in sequence. It is recommended to first present the antimicrobial resistance data, followed by the discussion of the antimicrobial resistance genes. Additionally, the MIC breakpoints and results for each antibiotic tested should be presented in a table format to better illustrate the findings and enhance the clarity and persuasiveness of the results. Responds: The heading and the section have been reorganized appropriately. Doc please do not know if this can be provided as the original manuscript table only showed Resistant/ sensitive without any values with no MIC data.
13. Line 289: The phrase “(selm, and seln)” contains "and" which may be unnecessary. Response: Done
14. Line 294: The phrase “resistant to only penicillin” should be revised to “resistant only to penicillin” for better flow and clarity. Response: Done
15. Line 308-312: The sentence could be reworded for clarity and flow. A suggestion might be: “For example, one MRSA isolate carried five resistance genes (ant(6)-Ia, aph(3')-III, blaZ, mph(C), and msr(A)) on a single plasmid of 26,974bp, which provides resistance to aminoglycosides, beta-lactams, and macrolides. Plasmids are recognized as mobile genetic elements in S. aureus, enabling it to rapidly adapt to the selective pressures imposed by humans.” Response: The reviewer’s suggestion has been adopted
16. line 324: It is recommended to standardize the term to “spa type” throughout the manuscript to avoid inconsistencies between “spa-type” and “spa type, ensuring consistency and professionalism in the text. Response: Done
Reviewer #2 Title: The title is clear and accurately reflects the content. To enhance specificity, please mention the location at the end of the title: “…….in Alabama, USA” Response: Done
Introduction: The aim is clearly stated and directly addresses a critical gap in the literature. However, the objectives could be more specific by detailing how findings will inform clinical or public health interventions. Response: This has been clarified
Methods, such as molecular techniques and susceptibility testing protocols, are clearly described, ensuring replicability.
Major concern: Sample size calculation. I have major concerns on the use of ref. 32 to do the calculation. Since no data is available on MDR S. aureus in the state of Alabama, it should be based on 50% prevalence as standard protocol for unknown prevalence calculation??? You cannot take 26% from other state to do the calculation. Please recalculate the sample size and do the study Or simply remove this section and remove the term PREVALENCE from title and other part. Response: The suggestion by the reviewer is deem appropriate and has therefore been adopted
Line 223, how the p-value was determined, write in methodology, please. Response: This was obtained by performing ANOVA statistics on the data using Graphpad Prism version 10.3. and it has been included in the methodology
Line 280, what are those resident microbes, name please? Response: Some of them have been named as requested.
The results are presented systematically, with clear tables and figures. Statistical analyses are appropriate for the data, and the findings are logically interpreted. However, there are not enough discussion on the implications or public health significance of the findings. Please add few more sentences in this regard. Response: Thanks for the suggestions we consider the current discussion to suffice for the information we wish to communicate.
Please write the conclusion under a separate heading Response: This has been done
The Figures are clear and effectively complement the text. However, Figure legends could provide more detail, especially for molecular results. Response: Thanks for the comments. Molecular results have been provided in the supplementary data.
Other comments: Please add plates of sensitivity test and PCR gel images in suppl. File Response: Thanks for pointing that out. Gel images have been supplied as part of the supplementary data and the MIC results obtained from the sensitivity analysis as well
The manuscript is well-written, with minimal grammatical issues. However, some corrections please as follows:
Line 294, S. aureus make Italic Response: Done
Line 302, emr italic pls. Response: Done
Reviewer #3 Line 1: Response: “Title” has been removed, the caption modified to reflect origin of samples and all Staphylococcus aureus have been italicized throughout the manuscript
Line 23: Please provide more details about the materials and methods. Number of samples, collection period, analyses performed, data processing, etc. Response: To stay within the word limit for the abstract, the requested details have been provided in the methodology.
Linie 25: Is positivity alone important for this type of food? Wouldn't it be more important to determine the quantity since it is expected that processed foods will have a certain level of S. aureus contamination. Response: While we recognize the importance of determining the quantity, it was beyond the scope of our study due to resource limitations. Our objective was solely to assess the frequency of pathogen isolation from the samples.
Line 28: Which and how many antibiotics were tested? Response: Again, to stay within the word limit for the abstract, the requested details have been provided in the methodology.
Line 69: What is the microbiological standard for S. aureus in this type of food? Cite methodology. Response: Thanks for the comment. The manuscript has been revised to provide this information, and this has been appropriately cited
Line 139: Please indicate statistical analyses in a separate item at the end of all analyses performed Response: All statistical analysis were performed using ANOVA in GraphPad Prism version 10.3. and it has been included in the methodology
Line 218: ... or frequency? Response: We agree with the reviewer’s suggestion and have adopted it appropriately as our estimation of the sample size coupled with limited resources will not allow us to perform an efficient prevalence analysis
Reviewer #4 Abstract: "the presence of antibiotic-resistant Staphylococcus aureus (S. aureus) in retail meat is a public health threat requiring continuous surveillance." Suggestion: "the presence of antibiotic-resistant Staphylococcus aureus (S. aureus) in retail meat poses a public health threat that requires continuous surveillance." Response: The Reviewer’s suggestion adopted
Introduction: "The U.S. is the largest producer and the second-largest exporter of poultry meat globally." Suggestion: "The U.S. is the largest producer and the second-largest exporter of poultry meat in the world." Response: The Reviewer’s suggestion adopted
Methodology: "Due to the lack of available research data on multidrug-resistant S. aureus in the state of Alabama, an assumed prevalence (Pexp) of 26% from chicken (breast & thighs) taken from a previous study was used to determine the sample size for this study." Suggestion: "Due to the lack of available research data on multidrug-resistant *S. aureus* in Alabama, an assumed prevalence (Pexp) of 26% from a previous study on chicken (breast and thighs) was used to determine the sample size. Response: The reviewer’s suggestion was implemented, but the corresponding manuscript section was removed as part of our proposed revisions. Since Reviewer 1 raised the same concern, this section has been excluded, and the term “prevalence” has been removed from the title as well.
Results: "The prevalence of S. aureus in the tested chicken meat products was higher in chicken thigh samples, 37.5% (12/32), followed by chicken wings at 34.4% (11/32)..." Suggestion: "The prevalence of S. aureus in the tested chicken meat products was highest in chicken thigh samples at 37.5% (12/32), followed by chicken wings at 34.4% (11/32)..." Response: The Reviewer’s suggestion adopted
Section 3.4: "The plasmid DNA from one of the isolates (spa-type t0080) was sequenced." Suggestion: "Plasmid DNA from one of the isolates (spa-type t0080) was sequenced." Response: The Reviewer’s suggestion adopted
Discussion: "This result is comparatively similar to another study in the U.S. which reported an S. aureus prevalence of 17.8% in retail chicken." Suggestion: "This result is comparable to another study in the U.S. that reported an S. aureus prevalence of 17.8% in retail chicken." "The toxins with the ability to induce a vomitory effect have sometimes specifically been referred to as SE..." Suggestion: "Toxins that induce a vomitory effect are sometimes referred to as SE..." Response: The Reviewer’s suggestion adopted
"The presence of resistant bacteria in chickens poses a significant risk to human health." Suggestion: "The presence of resistant bacteria in chickens poses a significant risk to human health." Response: The Reviewer’s suggestion adopted
"We infer that these genes, although present, were not phenotypically expressed." Suggestion: "We infer that these genes, while present, were not phenotypically expressed." Response: The Reviewer’s suggestion adopted
"A worth-noting finding was that almost all the antimicrobial genes (except mecA)..." Suggestion: "A noteworthy finding was that almost all the antimicrobial genes (except mecA)..." Response: The Reviewer’s suggestion adopted
"The finding of plasmid-associated toxin and antimicrobial resistance genes agrees with the study of McCarthy and Lindsay who sequenced and studied 253 S. aureus plasmids." Suggestion: "The finding of plasmid-associated toxin and antimicrobial resistance genes aligns with the study by McCarthy and Lindsay, who sequenced and analyzed 253 S. aureus plasmids." Response: Reviewer’s suggestion adopted
|
4. Response to Comments on the Quality of English Language |
Point 1: |
Response: We shall accept the English revision service of your journal
|
5. Additional clarifications |
Response: We wish to thank them once again for taking the time to review this manuscript. |

Round 2
Reviewer 2 Report
Comments and Suggestions for Authors
Thanks for the update,
Reviewer 3 Report
Comments and Suggestions for Authors
The authors corrected the manuscript.